# Evaluating the efficacy of serological testing of clinical specimens collected from patients with suspected brucellosis

**Nannan Xu**[1☯‡], **Chunmei Qu**[1☯‡], **Lintao Sai**[1], **Sai Wen**[1], **Lulu Yang**[1], **Shanshan Wang**[1], **Hui Yang**[1], **Hui Liu**[2]*, **Gang Wang**[1]*

**1** Department of Infectious Disease, Qilu Hospital, Cheeloo College of Medicine, Shandong University, Jinan Shandong, China, **2** Institute of Bacterial Disease, Jinan Center for Disease Control and Prevention, Jinan Shandong, China

☯ These authors contributed equally to this work.
‡ The authors are listed in order of increasing seniority.
* dahuiliu1981@sina.com (HL); wangg1975@hotmail.com (GW)

**Data Availability Statement:** All relevant data are within the manuscript and its Supporting Information files.

## Abstract

### Background

This study aims to evaluate the efficacy of the standard agglutination test (SAT), the Brucellacapt test and enzyme-linked immunosorbent assay (ELISA) in clinical specimens collected from patients with suspected brucellosis.

### Methods

A prospective study was conducted from December 2020 to December 2021. Brucellosis was diagnosed on the basis of clinical evidence, and confirmed by isolation of *Brucella* or a four-fold rise in SAT titer. All samples were tested by the SAT, ELISA and the Brucellacapt test. Titers ≥1:100 were considered as SAT positive; ELISA was considered positive when an index greater than 11 was detected, while titers ≥1/160 indicated positivity on the Brucellacapt test. The specificity, sensitivity, and positive (PPVs) and negative predictive values (NPVs) of the three different methods were calculated.

### Results

A total of 149 samples were collected from patients with suspected brucellosis. The sensitivities for the SAT, IgG, and IgM detection were 74.42%, 88.37% and 74.42%, respectively. The specificities were 95.24%, 93.65%, and 88.89%, respectively. The simultaneous measurement of IgG and IgM improved the sensitivity (98.84%) but reduced the specificity (84.13%) compared to each antibody test separately. The Brucellacapt test had excellent specificity (100%) and a high PPV (100%); however, the sensitivity and NPV were 88.37% and 86.30%, respectively. The combination of IgG detection by ELISA and the Brucellacapt test had excellent diagnostic performance, with 98.84% sensitivity and 93.65% specificity.

**Funding:** The author(s) received no specific funding for this work.

**Competing interests:** The authors declare that they have no conflicts of interest.

## Conclusion

This study showed that the simultaneous performance of IgG detection by ELISA and the Brucellacapt test has the potential to overcome the current limitations of detection.

Author summary

Brucellosis is an endemic zoonotic disease with significant impacts on public health worldwide. Early identification is crucial to reduce the rates of disability and mortality due to brucellosis. Human brucellosis is associated with a wide variety of clinical manifestations, making it difficult to diagnose clinically. Serological tests, which lack specificity and produce results that may be difficult to interpret, play a fundamental role in the diagnosis of this disease. Therefore, the combination of various serological tests may be helpful in ensuring the accuracy of diagnosis. Here, we aimed to explore the best combination among the SAT, ELISA and the Brucellacapt test to improve the efficiency of the diagnosis of human brucellosis.

## Introduction

Brucellosis is a zoonotic infectious disease caused by *Brucella* spp. [1]. To date, there are 12 species in the genus *Brucella* [2], among which *B. melitensis*, *B. abortus*, *B. suis*, and *B. canis* are pathogenic to humans [3]. More than 500,000 newly confirmed cases of human brucellosis occur worldwide every year [3]. In the past decade, the epidemiological characteristics of human brucellosis in China have changed substantially [4]. Brucellosis has gradually become one of the most prevalent infectious diseases and a serious public health threat [5]. This disease mainly causes losses in working time and an increased financial burden related to delayed diagnosis and a long treatment duration [6]. Brucellosis is easily mistaken for other medical conditions because of its variability and nonspecific clinical manifestations [7]. It is believed that the global incidence of brucellosis might be much higher than generally estimated [8,9]. Therefore, laboratory diagnosis is essential for proper treatment.

There are various assays available for the diagnosis of *Brucella* infection, including molecular, serological, and microbiological tests. Culture is the gold standard method for *Brucella* detection, but the sensitivity of blood culture had a wide range, from 10% to 90% [7]. Real-time polymerase chain reaction (PCR) is the most commonly used molecular method currently. However, the number of commercial PCR assays is limited, and there are significant differences among various commercial kits for the detection of *Brucella* DNA [10–12]. Because the clinical characteristics of human brucellosis are diverse and sample types are variable, further research is needed to determine the most effective detection protocol for each sample type. Serological tests, which lack specificity and produce results that may be difficult to interpret, play a fundamental role in the diagnosis of this disease [10,11,13].

The Brucellacapt test (Vircell SL) is a novel single-step immunocapture assay that has been applied in the serological diagnosis of human brucellosis, as it detects nonagglutinating IgG and IgA antibodies as well as agglutinating antibodies [14]. It has a higher sensitivity and specificity than agglutination tests as well as good correlation with the Coombs test [15,16]. In this study, we aimed to evaluate the value of commonly used serological tests, including the Brucellacapt test, enzyme-linked immunosorbent assay (ELISA) and the standard tube agglutination (SAT), in the diagnosis of human brucellosis in clinical practice.

## Materials and methods

### Ethics statement

This study was approved by the Shandong University Qilu Hospital human research protection committee (KYLL-202008-058). All patients signed consent forms.

### Serum sample collection

In this prospective study, we consecutively recruited 149 patients with suspected brucellosis from December 2020 to December 2021 from Qilu Hospital of Shandong University. These patients presented with symptoms such as headache, fever, chills, fatigue, joint pain, back pain and weight loss. Clinical and epidemiological information was gathered, and initial blood samples were obtained for clinical use. An additional venous blood sample was collected simultaneously from the patient. After 2–4 weeks, a second serum sample was collected to check for an increase in antibodies. In order to evaluate the background antibody titers in the normal population, we recruited 50 healthy adults for serological testing. The isolated serum was divided into aliquots and stored at −20°C until use.

The diagnosis of Brucellosis was based on the proper clinical context, including history (occupationally exposed or consumption of raw dairy/meat product or living in endemic areas), clinical presentation (fever, sweating, arthralgia, hepatosplenomegaly) and laboratory studies as well as at least one of the following results being positive: bacterial culture or four-fold or greater rise in SAT titer. Considering the time from onset to first admission, the duration of the disease was less than 8 weeks in the acute phase, 8–24 weeks in the subacute phase, and > 24 weeks in the chronic phase.

### SAT

The SAT antigen was purchased from the China Center for Disease Control and Prevention. Patient serum was serially diluted from 1/25 to 1/200 using phenol saline. *Brucella* antigen was added, and the mixture was incubated at 37°C for 24 h. The samples were examined for the presence of agglutinin particles. All tubes were compared with control tubes (positive and negative controls) to examine agglutination. Titers ≥1:100 with a minimum of 50% agglutination were considered positive.

### ELISA

Serum levels of anti-Brucella IgM and IgG were determined using ELISA kits according to the manufacturer's instructions (Vircell SL, Santa Fe, Granada, Spain) [17,18]. In brief, 5 µl of serum was added to a 100 µl of serum diluent in each microplate well. 25 µl of human IgG sorbent was included in the preparation of the serum diluent for the detection of only IgM antibodies. The microplates were covered with sealing mats and incubated at 37°C for 45 minutes. After washing 5 times with PBS, 100 µl of anti-human peroxidase conjugate IgG and IgM was applied to all microplate wells, and the plates were incubated at 37°C for 30 minutes. After a second wash, the substrate for the enzyme was added. After 20 minutes, stop buffer was added, and the absorbance was measured at 450 nm. In each run, positive, negative and cut off controls provided by Vircell for ELISA IgG and IgM were included. The qualitative Vircell ELISA IgM and IgG assays to detect positivity or negativity used a screening dilution of 1:20. The interpretation of results was carried out using an antibody index [(absorbance of the sample/ average absorbance of the serum cutoff value) × 10]. Samples with an index > 11 were considered positive. The combined results of IgG and IgM detection by ELISA were determined as

follows: when one of the ELISA results was positive, the case was considered positive, and when both of the ELISA results were negative, the case was considered negative.

## Brucellacapt test

The Brucellacapt test consists of microplates coated with total anti-human immunoglobulin. The Brucellacapt test was carried out according to the manufacturer's instructions (Vircell SL, Santa Fe, Granada, Spain) [19]. The diagnostic threshold titer for the Brucellacapt test was 1/160.

## Data analysis

Necessary data of patients, such as demographic, symptom, duration from symptom onset, and occupational risk factor data, were collected from the hospital database. Categorical variables are described using frequencies and percentages, while continuous variables are described using medians and interquartile ranges (IQRs). Baseline features were compared between two groups using Fisher's exact test (categorical variables), Student's test or the Mann–Whitney U test. The specificity, sensitivity, and positive (PPVs) and negative predictive values (NPVs) of the different methods were obtained with OpenEpi version 3.0 (http://www.openepi.com/). The 95% confidence intervals were calculated using Wilson's method. The receiver operator characteristic curve (ROC) was used to evaluate the performance of each method or combination. The cutoff value was derived using the Youden method [20]. All sensitivity and specificity differences between the tests were calculated using McNemar's test. Statistical analyses were conducted using SPSS version 23 (Inc., Chicago, IL, USA). A $P$ value $\leq 0.05$ was considered to be statistically significant.

This study was approved by the Institutional Ethics Committee of Shandong University Qilu hospital (KYLL-202008-058).

## Results

### Patient characteristics

Among 149 suspected brucellosis patients, 86 patients were diagnosed with brucellosis, of which 32 patients had positive culture results (the gold standard for the diagnosis of brucellosis); the diagnosis of the other 54 patients was based on epidemiological, clinical and serological criteria. Sixty-three patients had diseases other than brucellosis, including 40 cases of infectious diseases (bacterial infection in 34 cases, viral infection in 3 cases and fungal infection in 3 cases), 23 noninfectious diseases. Among the brucellosis patients, the median age was 54.5 (IQR: 41–64) years, and the numbers of females and males were 26 (30.2%) and 60 (69.8%), respectively. The group of patients with other diseases consisted of 27 females and 36 males, with a median age of 58 (IQR: 50–68) years. There was no significant difference between the two groups in age or sex. The most common symptoms in brucellosis patients were fever (82.6%), arthralgia (43.0%), fatigue (40.7%) and weight loss (27.9%), but there were no significant differences in symptoms compared with patients with other diseases. However, back pain (34.9% vs. 14.3%, p = 0.005) was reported in patients with brucellosis but not common in patients without brucellosis. Regarding laboratory findings, a significantly lower white blood cell count (P <0.001), C-reactive protein level (P = 0.028) and procalcitonin level (P = 0.002) were observed in patients with brucellosis than in patients without brucellosis. In contrast, hemoglobin levels in the brucellosis patients were higher than those in patients without brucellosis (P <0.001). Table 1 summarizes the demographic, clinical and laboratory data of these patients.

**Table 1. Demographic and clinical characteristics of the patients with brucellosis and diseases other than brucellosis.**

| Characteristics | Brucellosis (n = 86) | Other diseases (n = 63) | P value |
|---|---|---|---|
| Age, years | 54.5(41–64) | 58(50–68) | 0.058 |
| Gender | | | 0.112 |
| Male | 60(69.8%) | 36(57.1%) | |
| Female | 26(30.2%) | 27(42.9%) | |
| Duration of illness at diagnosis | | | 0.083 |
| < 8 weeks | 36(41.9%) | 38(60.3%) | |
| 8–24 weeks | 29(33.7%) | 15(23.8%) | |
| > 24 weeks | 21(24.4%) | 10(15.9%) | |
| Contact history | 40(46.5%) | 18(28.6%) | 0.027 |
| Clinical presentation | | | |
| Fever | 71(82.6%) | 57(90.5%) | 0.170 |
| Sweating | 14(16.3%) | 4(6.3%) | 0.066 |
| Arthralgia | 37(43.0%) | 33(52.4%) | 0.258 |
| Back pain | 30(34.9%) | 9(14.3%) | 0.005 |
| Fatigue | 35(40.7%) | 33(52.4%) | 0.157 |
| Headache | 22(25.6%) | 20(31.7%) | 0.409 |
| Hepatosplenomegaly | 16(18.6%) | 10(15.9%) | 0.664 |
| Lymphadenectasis | 16(18.6%) | 13(21.0%) | 0.757 |
| Weight loss | 24(27.9%) | 19(30.2%) | 0.764 |
| **Laboratory findings** | | | |
| White blood cell count, x$10^9$ /L | 6.05(4.55–8.17) | 8.16(6.07–11.56) | <0.001 |
| Neutrophils, % | 58.8(52.0–70.7) | 74.2(64.1–83.3) | <0.001 |
| Hemoglobin, g/L | 122(111–139) | 110(94–119) | <0.001 |
| Platelet count, x$10^9$/L | 238(192–316) | 272(189–392) | 0.160 |
| ALT, U/L | 27(16–45) | 28(16–49) | 0.869 |
| AST, U/L | 26(18–37) | 28(18–48) | 0.419 |
| Creatinine, μmol/L | 60(50–70) | 57(44–67) | 0.267 |
| BUN, mmol/L | 4.8(4.0–5.9) | 4.3(3.4–5.7) | 0.061 |
| ESR, mm/h | 35(16–70) | 71(36–105) | <0.001 |
| CRP, mg/L | 14.23(4.78–55.12) | 36.6(6.96–92.98) | 0.028 |
| PCT, ng/ml | 0.087(0.037–0.191) | 0.149(0.075–0.252) | 0.002 |

Reported counts (proportions) for categorical and median (interquartile range) for continuous variables. *P* values indicate differences between patients with brucellosis and patients with other diseases. *P* < 0.05 was considered statistically significant.

Abbreviations: ALT, alanine aminotransferase; AST, aspartate aminotransferase; BUN, blood urea nitrogen; ESR, erythrocyte sedimentation rate; CRP, C-reactive protein; PCT, procalcitonin

Upon admission, 36 (41.9%) patients presented with acute-stage disease, 29 (33.7%) patients presented with subacute-stage disease, and 21 (24.4%) patients presented with chronic-stage disease. In 86 patients with brucellosis, 47 (54.7%) had various focal complications, among which osteoarticular involvement was the most common. *Brucella* species from 32 patients were isolated from 31 blood cultures and 1 synovial fluid culture, and the positive rate of culture was 37.2%.

## Overall performance of the serological tests

**SAT and ELISA detection results.** Among the 86 samples from confirmed brucellosis patients, 64, 64 and 76 samples were determined to be positive on the SAT and by IgM and

**Table 2. The diagnostic performance of the SAT, ELISA and Brucellacapt.**

|  | Sensitivity (95% CI) | Specificity (95% CI) | PPA (95% CI) | NPA (95% CI) | Accuracy (95% CI) |
|---|---|---|---|---|---|
| SAT | 74.42% (64.29,82.46) | 95.24% (86.91, 98.37) | 95.52% (87.64, 98.47) | 73.17% (62.7, 81.56) | 83.22% (76.4, 88.37) |
| IgM | 74.42% (64.29,82.46) | 88.89% (78.8, 94.51) | 90.14% (81.02, 95.14) | 71.79% (60.97, 80.57) | 80.54% (73.45, 86.09) |
| IgG | 88.37% (79.9, 93.56) | 93.65% (84.78, 97.5) | 95% (87.84, 98.04) | 85.51% (75.34, 91.93) | 90.60% (84.85, 94.32) |
| Brucellacapt | 88.37% (79.9, 93.56) | 100% (94.25, 100) | 100% (95.19, 100) | 86.3% (76.59, 92.39) | 93.29% (88.09, 96.31) |
| IgM+ IgG | 98.84% (93.7, 99.79) | 84.13% (73.19, 91.14) | 89.47% (81.7, 94.18) | 98.15% (90.23, 99.67) | 92.62% (87.26, 95.83) |
| SAT+ Brucellacapt | 88.37% (79.9, 93.56) | 95.24% (86.91, 98.37) | 96.2% (89.42, 98.7) | 85.71% (75.66, 92.05) | 91.28% (85.65, 94.83) |
| IgM+ Brucellacapt | 93.02% (85.6, 96.76) | 88.89% (78.8, 94.51) | 91.95% (84.31, 96.05) | 90.32% (80.45, 95.49) | 91.28% (85.65, 94.83) |
| IgG+ Brucellacapt | 98.84% (93.7, 99.79) | 93.65% (84.78, 97.5) | 95.51% (89.01, 98.24) | 98.33% (91.14, 99.71) | 96.64% (92.39, 98.56) |
| IgM+ IgG + Brucellacapt | 100% (95.72, 100) | 84.13% (73.19, 91.14) | 89.58% (81.88, 94.24) | 100% (93.24, 100) | 93.29% (88.09, 96.31) |

Titers ≥1:100 were considered as SAT positive; IgG or IgM was considered positive when an index greater than 11 was detected, while titers of 1/160 and above indicated positivity on the Brucellacapt test.

The combined serological tests results were determined as follows: when one of the results was positive, the case was considered positive, and when all the results were negative, the case was considered negative.

Abbreviations: SAT, standard tube agglutination; IgM, IgM antibodies detection by ELISA; IgG, IgG antibodies detection by ELISA; PPA, positive predictive value; NPA, negative predictive value; 95% CI, 95% confidence intervals.

IgG detection by ELISA, respectively. Eighty-five samples were positive according to the combined ELISA (IgG + IgM), see S1 Material for details. Among the patients without brucellosis, the SAT results were positive in 3 cases and negative in 60 cases; the IgM results were positive in 7 cases and negative in 56 cases; and the IgG results were positive in 4 cases and negative in 59 cases. For the combined ELISA (IgG + IgM), the results were positive in 10 cases and negative in 53 cases. Among 50 healthy adults, only one was IgM positive, but both SAT and IgG were negative. Detailed serological results were shown in S2 Material.

The sensitivity values for the SAT, IgG detection, and IgM detection were 74.42% (95% CI 64.29, 82.46), 88.37% (95% CI 79.90, 93.56) and 74.42% (95% CI 64.29, 82.46), respectively. The specificity values for the SAT, IgG detection, and IgM detection were 95.24% (95% CI 86.91, 98.37), 93.65% (95% CI 84.78, 97.50), and 88.89% (95% CI 78.8, 94.51), respectively. The overall diagnostic performance of the SAT and ELISA for the diagnosis of brucellosis is summarized in Table 2. Compared with the detection of IgG and IgM separately, the combined ELISA results had a sensitivity of 98.84% (95% CI 93.70, 99.79) for the detection of brucellosis (p = 0.004 and p <0.001). However, the specificity decreased to 84.13% (95% CI 73.19, 91.14), which was lower than that for the IgG ELISA (p = 0.031).

## Brucellacapt detection results

The analysis of the Brucellacapt test results of the 149 serum samples revealed that 73 samples were classified as negative, 6 samples were 1/160 positive, 8 samples were 1/320 positive, and 62 samples were ≥1/640 positive. As expected, none of the 63 patients without brucellosis were positive on the Brucellacapt test. Interestingly, 10 samples with positive results according to the ELISA were negative on the Brucellacapt test. No healthy individuals had Brucellacapt titers of ≥ 1:160. With a titer of 1/160 as the threshold level for positivity, the Brucellacapt test had a sensitivity of 88.37% (95% CI 79.90, 93.56) and a specificity of 100% (95% CI 94.25, 100). The PPV was 100% (95% CI 95.19, 100), and the NPV was 86.30% (95% CI 76.59, 92.39). The overall diagnostic accuracy was 93.29% (95% CI 88.09, 96.31). However, when titers of 1/320 and higher were considered positive, we found that the sensitivity and NPV of the Brucellacapt test decreased to 81.40% (95% CI 71.89, 88.21) and 79.75% (95% CI 69.6, 87.13), respectively,

while the specificity and PPV remained unchanged. In order to obtain greater efficiency, we adjusted the cutoff value according to the ROC curve, the optimal cutoff value was ≥ 1:160. For details, please see the S3 Material. Therefore, values of 1/160 and higher were considered positive for the Brucellacapt test and used for subsequent analysis.

The Brucellacapt test and IgG detection by ELISA had similar sensitivities, while IgM detection by ELISA and the SAT had lower sensitivities than the Brucellacapt test (p = 0.012 and p <0.001, respectively). The Brucellacapt test had a higher specificity than IgM detection by ELISA (p = 0.016). However, no significant differences in the specificity between the Brucellacapt test, IgG detection and the SAT were found.

## Comparative analysis of serological tests

While culture is considered the gold standard, the SAT was positive in 26 (81.25%) patients, IgM detection by ELISA was positive in 27 (84.38%) patients, IgG detection by ELISA was positive in 23 (71.88%) patients and the Brucellacapt test was positive in 30 (93.75%) patients (Table 3). The Brucellacapt test had a similar sensitivity to IgM detection and the SAT but a higher sensitivity than IgG detection (p = 0.039). When IgM detection was combined with IgG detection, ELISA (IgG + IgM) had a similar sensitivity to the Brucellacapt test.

We subsequently combined different IgM, IgG and Brucellacapt tests to identify the best combination for the diagnosis of human brucellosis. As shown in Table 2, ELISA (IgG + IgM) was found to be more sensitive than the Brucellacapt test in detecting brucellosis (P = 0.012), whereas the Brucellacapt test had a higher specificity (100%) (P = 0.002). Considering the sensitivity and specificity, the combination of IgG detection by ELISA and the Brucellacapt test had excellent diagnostic efficacy, with 98.84% sensitivity and 93.65% specificity.

## Discussion

*Brucella* infection remains endemic in northeast China, and the incidence of brucellosis has increased in recent years [21,22]. Brucellosis is treatable, but this infection can lead to a severe and prolonged illness in humans in certain cases [6,23]. Early and reliable diagnosis followed by appropriate antibiotic treatment is crucial, thus preventing chronic disease and focal complications [24,25]. Although the interpretation of serological tests is usually difficult, especially in patients with chronic brucellosis, reinfection or recurrence and those in epidemic areas, serological methods play a key role in the routine diagnosis of brucellosis [10]. To overcome limitations of serological tests used to diagnose brucellosis, the combination of various serological tests, including different test methods, may be helpful in ensuring quality.

The most popular serologic tests for the diagnosis of human brucellosis are the SAT, the Rose Bengal test (RBT), the Coombs test and ELISA. According to their overall accuracy in

**Table 3. Results of the culture, SAT, ELISA and Brucellacapt performed on 86 brucellosis patients.**

| Cases | SAT: n (%) | | IgM: n (%) | | IgG: n (%) | | Brucellacapt: n (%) | |
|---|---|---|---|---|---|---|---|---|
| | Positive | Negative | Positive | Negative | Positive | Negative | Positive | Negative |
| Total (N = 86) | 64(74.42) | 22(25.58) | 85(98.84) | 1(1.16) | 85(98.84) | 1(1.16) | 76(88.37) | 10(11.63) |
| Culture positive (n = 32) | 26(81.25) | 6(18.75) | 27(84.38) | 5(15.62) | 23(71.88) | 9(28.12) | 30(93.75) | 2(6.25) |
| Culture negative (n = 54) | 38(70.37) | 16(29.63) | 37(68.52) | 17(31.48) | 53(98.15) | 1(1.85) | 46(85.19) | 8(14.81) |

Titers ≥1:100 were considered as SAT positive; IgG or IgM was considered positive when an index greater than 11 was detected, while titers of 1/160 and above indicated positivity on the Brucellacapt test.

Abbreviations: SAT, standard tube agglutination, IgM, IgM antibodies detection by ELISA; IgG, IgG antibodies detection by ELISA.

clinical settings, these test systems can be ranked as follows: ELISA > RBT > SAT > Coombs test [13].

ELISA can be used to reliably diagnose human brucellosis and is more sensitive than the SAT and RBT [14,26,27]. However, the sensitivity and specificity of ELISA for the detection of antibodies against *Brucella* spp. differ among studies. Araj GF et al. [27] compared ELISA with the SAT and Coombs test, and the sensitivities of IgG and IgM detection by ELISA were 91% and 100%, respectively, while the specificity was 100% for both. In contrast, in a study conducted by Memish et al. [28], the sensitivity and specificity were 45.5% and 97.1% for IgM and 79% and 100% for IgG, respectively; however, when the two ELISA results were evaluated together, the sensitivity and specificity were 94.1% and 97.1%, respectively. In our study, the sensitivity (88.37%) of IgG detection was higher than that of both IgM detection (74.42%) and the SAT (74.42%). We found that combined IgG and IgM results significantly improved the sensitivity (98.84%) but decreased the specificity (84.13%), which was similar to the results of previous studies [12,26]. Therefore, ELISA can diagnose human brucellosis with high sensitivity [13] but may not have sufficient specificity to be used as a diagnostic tool [29].

With the progression of the disease, IgG agglutinating antibodies gradually shift to nonagglutinating IgG antibodies [30]. Coombs test is necessary for the identification of blocking antibodies in the serologic diagnosis of *Brucella* infection. However, this kind of experiment is not routinely carried out in many clinical laboratories because it is technically difficult and requires skilled personnel.

The Brucellacapt test detects agglutinating and nonagglutinating antibodies [14] and is thus suggested to be a possible substitute for the Coombs test [14,15]. In accordance with these findings [31,32], the results of our study also clearly demonstrated that the Brucellacapt test had higher specificity in the diagnosis of human brucellosis. In contrast, Ardic et al. [33] found that the sensitivity, specificity, PPV and NPV of the Brucellacapt test (1/160) were 97.3%, 55.6%, 90% and 83.3%, respectively. The results may be related to the phase of *Brucella* infection. Another important observation from this study is that the best cutoff titer for the Brucellacapt test is ≥1:160, not 1:320. When titers ≥1/320 was used as a diagnostic threshold, the Brucellacapt test sensitivity decreased slightly. In addition, the Brucellacapt test was previously reported to have good sensitivity and specificity in the diagnosis of chronic brucellosis [14]. However, negative Brucellacapt results were observed in chronic brucellosis patients in this study.

For reliable serological diagnosis of human brucellosis, at least two different tests are needed: one based on a high-sensitivity screening method, and another based on more specific methods to confirm the preliminary test results. In addition, from a clinician's perspective, the predictive value or probability of disease given a test result is the most important aspect of test performance [34]. In our study, ELISA had a high sensitivity (98.84%) and NPV (98.15%), making it a very useful tool for the rapid screening of endemic populations. The Brucellacapt test has excellent specificity and a high PPV (exceeding 100%), allowing the determination of whether the patient is truly positive for brucellosis. Taken together, the combination of IgG detection by ELISA and the Brucellacapt test seems to have the best sensitivity and specificity compared to the other tests used in this study. The remarkable sensitivity and specificity make IgG detection by ELISA combined with the Brucellacapt test very useful for the diagnosis of the disease in suspected patients and for the rapid screening of endemic populations.

A limitation of this study is that only 149 serum samples were tested. In the future, more samples from multiple centers need to be tested to determine the sensitivity and specificity in various contexts and patient types. There is no perfect diagnosis reference for Brucella infection, so using culture (or dynamic increases in SAT titers) as a reference, estimates of sensitivity and specificity will always be biased. Additionally, the combined tests need to be tested in epidemic regions to assess their value as a practical initial diagnostic test for brucellosis.

## Conclusion

Diagnosis of Brucellosis in humans is still a great challenge. Serological diagnosis of human brucellosis is imperfect but essential. Based on the results of this study, we believe that the combined use of IgG detection by ELISA and the Brucellacapt test has presented greater capacity for positive and negative classification of brucellosis in suspected patients. It can promote the correct identification of cases, improve and standardize the clinical management of brucellosis.

## Supporting information

**S1 Data. Supporting data for Table 1.**
(XLSX)

**S2 Data. Supporting data for Table 2.**
(XLSX)

**S3 Data. Supporting data for Table 3.**
(XLSX)

**S1 Material. Serological and Brucella culture results of all patients.**
(XLSX)

**S2 Material. Results of SAT, ELISA and Brucellacapt tests performed on 50 healthy individuals.**
(XLSX)

**S3 Material. Supplementary statistical analysis and results.**
(DOCX)

## Author Contributions

**Conceptualization:** Gang Wang.

**Data curation:** Chunmei Qu.

**Formal analysis:** Nannan Xu.

**Investigation:** Chunmei Qu, Lintao Sai, Sai Wen, Lulu Yang, Shanshan Wang, Hui Yang.

**Resources:** Hui Liu.

**Writing – original draft:** Nannan Xu.

**Writing – review & editing:** Hui Liu, Gang Wang.

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
