## [Decision Letter · Decision Letter 0]

10 Aug 2022

Dear Dr Wang

Thank you very much for submitting your manuscript "Evaluating the efficacy of serological testing of clinical specimens collected from patients with suspected brucellosis" for consideration at PLOS Neglected Tropical Diseases. As with all papers reviewed by the journal, your manuscript was reviewed by members of the editorial board and by several independent reviewers. In light of the reviews (below this email), we would like to invite the resubmission of a significantly-revised version that takes into account the reviewers' comments. 

We cannot make any decision about publication until we have seen the revised manuscript and your response to the reviewers' comments. Your revised manuscript is also likely to be sent to reviewers for further evaluation.

Sincerely,

David O'Callaghan

Guest Editor

Javier Pizarro-Cerda

Section Editor

 The reviewers feel that it is an interesting report. However, from their comments below, you will see that both have raised several points that need clarification and improvement.

These include a clarification over the study design, the definition of a positive brucellosis case and the use of appropriate statistical tests and reevaluation of the data.

Further to the reviewers comments, I have a few points that should also be addressed

The supplementary table is not very useful in its present form. First, the serological results should be given in the table for all the patients, second, the patients where a Brucella strain was isolated should be identified. Finally, rather than having the patients listed by number, please list them by duration of illness at diagnosis. 

L123. you read the Absorbance A450, not optical density, Same for line 127

There were 31 positive blood cultures and one synovial fluid. Does this represent 32 patients, or were there multiple positive samples form patients?

L156, what is a case of 'nonbrucellosis;? This is not a disease, please reword. As you use these patients as a control group, please describe which infections these patients had. this is important to rule out the possibility of cross reactivity.

I am in full agreement with reviewer 1, that it is ESSENTIAL to include an age matched healthy control group. This is also essential when describing the changes in blood biochemistry,

L 223 -6 and 299-9 The manufacturers instructions say that the limit of positivity with BrucellaCapt is 1/160. This is obviously based on extensive testing, what is the rational to alter the criteria to 1/320? It is evident that this will decrease the sensitivity and NPV. 

On the same line, you define 1/100 as positive with the SAT. I know that this is the titer defined by Chinese Law, but it is different from the recommendations in other countries (1/80; 120IU). How does difference effect your analysis? Please also indicate which positive serum was used and whether it was calibrated?

L301-2 You comment that some chronic patients were negative with BrucellaCapt. Please support this by giving the full serological and culture results of these patients in a table rather than leaving the reader to find it in the supplemental table

David O'Callaghan (Guest Editor)

Reviewer's Responses to Questions

**Key Review Criteria Required for Acceptance?**

**Methods**

-Are the objectives of the study clearly articulated with a clear testable hypothesis stated?

-Is the study design appropriate to address the stated objectives?

-Is the population clearly described and appropriate for the hypothesis being tested?

-Is the sample size sufficient to ensure adequate power to address the hypothesis being tested?

-Were correct statistical analysis used to support conclusions?

-Are there concerns about ethical or regulatory requirements being met?

Reviewer #1: The methods are clear although the criterion used to select the Brucellosis infection samples should be clarified and a true group of negative samples tested (see comments to authors).

Reviewer #2: McNamar’s test - please check the spelling

Was the study multicentric? or conducted in a single hospital? The information is not clear

How the specificity, sensitivity, and positive (PPVs) and negative predictive values (NPVs) were calculated? Using a frequentist approach I presume. It should be clearly explained in M&M and the authors should provide a reference for the calculation.

Were confidence interval calculated? How?

**Results**

-Does the analysis presented match the analysis plan?

-Are the results clearly and completely presented?

-Are the figures (Tables, Images) of sufficient quality for clarity?

Reviewer #1: The results are simple and clear but they should be reinterpreted after it is clarified how the positive samples were selected and, if necessary, modified accordingly. The same when a true negative group of samples are evaluated.

Reviewer #2: The other criteria used for the classification of cases without isolation should be clearly mentioned. How many patients were PCR positive among the cases for example?

The univariate analysis of the variables possibly associated with brucellosis is poorly suited to the inference of risk factors as it does not consider confounding factors, collinearity, etc. A multivariate model would be more suitable and informative.

How the results of different tests were combined? Considering a parallel testing? In this case how the final sensitivity and specificity were calculated? Did the authors considered that the results of the test are not indenpedent? Was the covariance between the tests estimated and considered in the calculation of combined Se and Sp? It must

Figure would be of great help to understand the data

**Conclusions**

-Are the conclusions supported by the data presented?

-Are the limitations of analysis clearly described?

-Do the authors discuss how these data can be helpful to advance our understanding of the topic under study?

-Is public health relevance addressed?

Reviewer #1: As with the results conclusions should be reevaluated once it is clarified how the positive samples were selected and, if necessary, modified accordingly. The same when a true negative group of samples are evaluated.

Reviewer #2: The analysis should be reviewed considering that the tests are not independent (they are all measuring anti-brucella antibodies)

**Editorial and Data Presentation Modifications?**

Reviewer #1: (No Response)

Reviewer #2: Major revision

**Summary and General Comments**

Reviewer #1: The paper by Xu et. al. describes the evaluation of several serological tests for the diagnosis of human brucellosis in patients with a potential active infection. The authors recruited 149 patients with a febrile illness compatible with brucellosis and evaluated five different techniques: blood culture, SAT, ELISA (IgG and IgM) and Brucellacapt. Using these tests, either individually or in combination, the authors propose the better combination that maximizes sensitivity and specificity. 

The manuscript is clear in the goals and the methods, and the conclusions are simple. Despite this a couple of things are not clear and should be clarified and modified.

1- What it is not clear is what criteria was used to diagnose brucellosis on the 149 patients. They state that a patient was diagnosed with brucellosis if it was blood culture positive or had a positive SAT. With this criterion 86 patients out of the 149 were included as positives. 32 were blood culture positive and 26 of these had SAT positive implying that 5 were negative which would mean that, out of the 86 considered as Brucellosis positive, 81 should be SAT positive. This is not what Table 2 shows. There it is stated that 64 of the 86 samples were SAT positive. This is all very confusing, if the criteria to diagnose brucellosis was SAT positive how is it possible that the sensitivity of the method is after this selection was only 74%? The correct way to evaluate this is to take one test as the gold standard and then compare the other ones against this group. As it is described, the analysis is very confusing.

2- The negative samples also have problems as all these samples come from potential infections (symptomatic). A true negative control group should be sera from healthy individuals.

Reviewer #2: it is an interesting article as it uses prospective sampling to assess the diagnostic sensitivity and specificity of serological tests for human brucellosis. In this sense, as it uses the same tests for definition of cases and non-cases, it has greater reilability of estimation than studies conducted in the case control format. However, the statistical approach used for the analysis is unclear and seems inadequate.

PLOS authors have the option to publish the peer review history of their article (what does this mean?). If published, this will include your full peer review and any attached files.

Reviewer #1: No

Reviewer #2: No
---

## [Decision Letter · Decision Letter 1]

17 Nov 2022

Dear Dr Wang.

Thank you for submitting a revised manuscript. You will see that both reviewers are happy with the way that you have addressed the majority of their comments.

However, Reviewer 2 still has major concerns over the statistical analysis used in the study. I invite you to address these concerns in a revised manuscript.

We look forwards to receiving a revised manuscript.

Best wishes

Dr David O'Callaghan. Guest Editor.

We cannot make any decision about publication until we have seen the revised manuscript and your response to the reviewers' comments. Your revised manuscript is also likely to be sent to reviewers for further evaluation.

Sincerely,

David O'Callaghan

Guest Editor

Javier Pizarro-Cerda

Section Editor

Dear Dr Wang.

Thank you for submitting a revised manuscript. You will see that both reviewers are happy with the way that you have addressed the majority of their comments.

However, Reviewer 2 still has major concerns over the statistical analysis used in the study. I invite you to address these concerns in a revised manuscript.

We look forwards to receiving a revised manuscript.

Best wishes

Dr David O'Callaghan. Guest Editor.

Reviewer's Responses to Questions

**Key Review Criteria Required for Acceptance?**

**Methods**

-Are the objectives of the study clearly articulated with a clear testable hypothesis stated?

-Is the study design appropriate to address the stated objectives?

-Is the population clearly described and appropriate for the hypothesis being tested?

-Is the sample size sufficient to ensure adequate power to address the hypothesis being tested?

-Were correct statistical analysis used to support conclusions?

-Are there concerns about ethical or regulatory requirements being met?

Reviewer #1: (No Response)

Reviewer #2: the "multivariate" statistical model used was not properly described, neither a description of the model nor of the references used. In addition, in the combinations of tests evaluated, the tests were considered independent to calculate Se and Sp, which is not true (all are basically measuring the same thing), the probability of the occurrence of one result is associated with the probability of another test. Thus, it would be necessary to evaluate the covariance between them and consider the value in the simulations of use in parallel.

**Results**

-Does the analysis presented match the analysis plan?

-Are the results clearly and completely presented?

-Are the figures (Tables, Images) of sufficient quality for clarity?

Reviewer #1: (No Response)

Reviewer #2: biased based on the statistical approach used

**Conclusions**

-Are the conclusions supported by the data presented?

-Are the limitations of analysis clearly described?

-Do the authors discuss how these data can be helpful to advance our understanding of the topic under study?

-Is public health relevance addressed?

Reviewer #1: (No Response)

Reviewer #2: either

**Editorial and Data Presentation Modifications?**

Reviewer #1: (No Response)

Reviewer #2: (No Response)

**Summary and General Comments**

Reviewer #1: The authors have addressed my questions and concerns.

Reviewer #2: (No Response)

PLOS authors have the option to publish the peer review history of their article (what does this mean?). If published, this will include your full peer review and any attached files.

Reviewer #1: No

Reviewer #2: No
---

## [Decision Letter · Decision Letter 2]

30 Jan 2023

Dear doctor gang,

We are pleased to inform you that your manuscript 'Evaluating the efficacy of serological testing of clinical specimens collected from patients with suspected brucellosis' has been provisionally accepted for publication in PLOS Neglected Tropical Diseases.

Best regards,

David O'Callaghan

Guest Editor

Javier Pizarro-Cerda

Academic Editor

<style type="text/css">p.p1 {margin: 0.0px 0.0px 0.0px 0.0px; line-height: 16.0px; font: 14.0px Arial; color: #323333; -webkit-text-stroke: #323333}span.s1 {font-kerning: none

</style>

Reviewer's Responses to Questions

**Key Review Criteria Required for Acceptance?**

**Methods**

-Are the objectives of the study clearly articulated with a clear testable hypothesis stated?

-Is the study design appropriate to address the stated objectives?

-Is the population clearly described and appropriate for the hypothesis being tested?

-Is the sample size sufficient to ensure adequate power to address the hypothesis being tested?

-Were correct statistical analysis used to support conclusions?

-Are there concerns about ethical or regulatory requirements being met?

Reviewer #1: (No Response)

Reviewer #2: (No Response)

**Results**

-Does the analysis presented match the analysis plan?

-Are the results clearly and completely presented?

-Are the figures (Tables, Images) of sufficient quality for clarity?

Reviewer #1: (No Response)

Reviewer #2: (No Response)

**Conclusions**

-Are the conclusions supported by the data presented?

-Are the limitations of analysis clearly described?

-Do the authors discuss how these data can be helpful to advance our understanding of the topic under study?

-Is public health relevance addressed?

Reviewer #1: (No Response)

Reviewer #2: (No Response)

**Editorial and Data Presentation Modifications?**

Reviewer #1: (No Response)

Reviewer #2: (No Response)

**Summary and General Comments**

Reviewer #1: (No Response)

Reviewer #2: (No Response)

PLOS authors have the option to publish the peer review history of their article (what does this mean?). If published, this will include your full peer review and any attached files.

Reviewer #1: No

Reviewer #2: No

---

## [Editor Report · Acceptance letter]

6 Feb 2023

Dear doctor Wang,

We are delighted to inform you that your manuscript, "Evaluating the efficacy of serological testing of clinical specimens collected from patients with suspected brucellosis," has been formally accepted for publication in PLOS Neglected Tropical Diseases.

Best regards,

Shaden Kamhawi

co-Editor-in-Chief

Paul Brindley

co-Editor-in-Chief
